# A Discrete Macro Element Method for Modelling Ductile Steel Frames around the Openings of URM Buildings as Low Impact Retrofitting Strategy

**Giuseppe Occhipinti** [1,*], **Francesco Cannizzaro** [2], **Salvatore Caddemi** [2] **and Ivo Caliò** [2]

1 Institute of Environmental Geology and Geoengineering, Italian National Research Council (CNR), Area della Ricerca di Roma 1, Strada Provinciale 35d, 00010 Rome, Italy
2 Department of Civil Engineering and Architecture, University of Catania, Viale A. Doria 6, 95131 Catania, Italy; francesco.cannizzaro@unict.it (F.C.); salvatore.caddemi@unict.it (S.C.); ivo.calio@unict.it (I.C.)
* Correspondence: giuseppe.occhipinti@igag.cnr.it

**Abstract:** This paper adopts the use of steel frames around existing openings as a low-impact seismic retrofitting strategy for unreinforced masonry structures (URM). Although elastic steel frames have been commonly adopted for strengthening masonry walls in case of the realization of new openings, the use of elasto-plastic frames has been proposed only recently. This study adopts the application of low-resistance ductile steel frames on the openings of existing masonry buildings as a low-impact retrofitting strategy. The adopted low-invasive solution possesses the advantage of increasing the in-plane resistance of the masonry wall, improving the displacement capacity, introducing additional energy dissipation under dynamic loadings, and providing a confinement effect on the adjacent masonry piers. An original aspect of the present paper is related to the adopted numerical method for modelling the presence of the steel frames around the openings. Namely, a Discrete Macro-Element Method (DMEM), which allows an efficient and reliable simulation of the involved collapse mechanisms of the masonry walls interacting with the frames, has been adopted. After the validation of the numerical approach, through a comparison with experimental results already reported in the literature, the low-impact strategy has been applied on a benchmark known as the "via Martoglio building". The obtained results suggest that this low-impact retrofitting strategy can be successfully proposed for URM buildings and can be efficiently modelled by means of the DMEM.

**Keywords:** Discrete Macro-Element-Method (DMEM); ductile steel frame; seismic retrofitting; low-impact retrofitting strategy; seismic vulnerability; 3DMacro

## 1. Introduction

Among the retrofitting strategies of existing masonry buildings, some of them are characterized by a limited interference with the activities that take place in the structure, which is a desirable condition to avoid relocation, even temporary, and to reduce the downtime period. Low-impact retrofitting strategies are usually characterized by fast application and limited cost, although leading to a significant reduction of the seismic vulnerability being conceived to act on specific collapse mechanisms by limiting the activation or increasing their resistance and ductility. A well-known example of low-invasive compatible retrofitting measure for masonry structure is represented by the tie-rods and plates whose adoption was mostly used in the past to eliminate the horizontal thrust of arches, vaults, and roofs. More recently many research groups have been involved in the definition of new low-invasive retrofitting strategies to be applied for seismic retrofitting in existing buildings (WP5 DPC-ReLUIS 2019-2021). However, their engineering and technological efficacy needs to be properly assessed and their practical applications should be evaluated and, when possible, guided by means of technical codes and recommendations and supported by suitable numerical methods.

A wide range of strengthening strategies has been proposed and analysed in the last several years [1] for masonry structures. A significant strengthening strategy of URM structures, which has gained a limited attention in the last several years, is based on the application of steel frames along the perimeter of openings. Traditionally, such a technique was widely employed to restore the original in-plane stiffness and resistance of masonry walls where new openings had to be realized [2]. Alternatively, other strategies aim at replacing entire masonry walls with frames endowed with the same in-plane horizontal stiffness and resistance [3]. The retrofitting strategy here recommended was recently applied for the structural retrofitting of an existing masonry school building according to a structural design based on a synergic combination of Glass Fiber Reinforced Cementitious Matrix composites and dissipative steel frames around the openings (*Progetto DPC-ReLUIS 2019-2021WP5: Interventi di rapida esecuzione a basso impatto ed integrati Case Study: SCUOLA IPSIA Vittoria (RG)*, in Italian). More recently Billi et al. [4] proposed design methods and performed nonlinear finite element simulations for forming new steel-framed openings in load-bearing masonry walls. The masonry wall is modelled through a rotating crack constitutive model, and the interaction between steel frame and masonry through cohesive-frictional interface elements. The influence of the steel-profile cross-section, the position of the opening within the wall, and the degree of connection between the steel uprights and the masonry are evaluated. Segovia-Verjel et al. [5,6] suggested a cost-effective retrofitting technique for URM buildings based on steel encirclements in existing openings. They evaluated the influence of the steel encirclements by using a frame modelling approach for the masonry building, without considering the complex interaction between the masonry and the frame that can modify the failure mechanism of the combined system and introduces an additional confinement effect both on the piers and the spandrels. Jorge Miguel Proença et al. [7] proposed a structural window frame for in-plane seismic strengthening of masonry wall buildings. The solution implemented ultimately aims to stiffen (and strengthen) the opening such that the wall would behave as if there were no opening. They performed an experimental test on a simple specimen with a central opening and simulated the outcome results by means of a nonlinear FEM strategy in which masonry has been modelled with a rotating crack model.

As already demonstrated by experimental and numerical results, the application of steel frames along the perimeter of existing openings can represent an efficacious and fast intervention that can also be easily combined with the replacement of the window frame for energy efficiency purposes. The steel frame can be designed in order to adhere to the perimeter of the opening and can be connected with the existing masonry by means of few dowels in order to constrain the system to be anchored in the plane; furthermore, the strength of the steel frame can be designed according to different purposes.

However, a reliable evaluation of this retrofitting strategy requires a numerical simulation able to account for the nonlinear coupling between the masonry and the surrounding steel frame. The numerical modelling of this strengthening technique is far from being easy and rigorously requires complex nonlinear FEM simulations, such as those adopted for simulating experimental results on prototypes in [4,7]. Oversimplified approaches, such those based on equivalent frame modelling, as those adopted in [5,6], are not able to provide a reliable simulation of the interaction between the steel frame and the masonry wall. It is worth noting that the introduction of an interacting steel frame contouring an existing opening is not a mere addition of stiffness and resistance for the structure. The presence of a frame around an opening plays a confinement effect on the surrounding piers and spandrels that can modify the expected damage mechanism, increasing the overall displacement capacity of the combined system. For the latter reason, Equivalent Frame Model (EFM) modelling strategies [8–10], which are widely adopted by practitioners for assessing the seismic performance of existing masonry buildings, do not appear effective for the simulation of this efficacious retrofitting methodology since they are based on uniaxial elements that cannot take into account the interaction between adjacent elements. On the other hand, rigorous nonlinear Finite Element Models (FEM), based on the spatiality

of the numerical models, although reliable, cannot be easily applied at the building scale in practical engineering due to the needed computational burden.

In this paper, a Discrete Macro-Element Model (DMEM) approach [11] is employed as a fair compromise between the required reliability and the possibility to adopt a model that can be employed by practical engineers for assessing the effectiveness of retrofitting strategies on existing masonry buildings. The considered approach is based on a bi-dimensional mechanical scheme that proved to be effective also in the case of mixed masonry-reinforced concrete structures [12–14]. The DMEM allows for effectively grasping the continuous interaction between masonry walls and beams, and its application for modelling URM buildings reinforced with steel frames at the perimeter of openings appears to be appropriate.

Aiming at validating the adopted approach against the application of steel frames around openings, the experimental tests on a single bay single story masonry wall, already presented in the literature [2], is first simulated. Then, a more general application on real-scale structures, the *via Martoglio wall* benchmark (which was the object of studies and simulations in the unreinforced condition [15]) is investigated by considering the openings encirclement as a possible low-impact and efficient retrofitting strategy. This study aims at showing the possibility of applying and efficiently numerically simulating steel frames around openings as a fast and structurally effective strategy for the seismic retrofitting of existing masonry buildings that can also be adopted in synergy with other cost-effective techniques.

## 2. The Discrete Macro-Element Model Approach

The Discrete Macro-Element Method (DMEM) [11] is a modelling strategy based on a simple mechanical scheme that was originally introduced to simulate the in-plane nonlinear behaviour of masonry walls. It belongs to the framework of simplified approaches being based on a discrete element at the macro-scale characterized by a very low computational burden [16]. Differently from the macro-element strategies based on the equivalent frame model, this numerical strategy is based on a geometrically consistent bi-dimensional scheme able to simulate the main in-plane collapse mechanisms of a masonry wall subjected to vertical and horizontal loads, namely rocking diagonal cracking and sliding failure modes (Figure 1).

The basic element is rectangular with rigid edges and hinges at the four corners; two diagonal nonlinear links govern the diagonal shear behaviour (Figure 2a), also accounting for the confinement effect associated with the interaction with the adjacent macro-elements or beams [17].

Nonlinear discretely distributed interfaces govern the interaction between contiguous elements, by means of nonlinear links whose calibration is based on a straightforward fiber approach, as qualitatively depicted in Figure 2b. A non-symmetric elasto-plastic behaviour with limited ductility is usually adopted and even considering the simplest of the nonlinear constitutive laws, namely an elastic-perfectly plastic behaviour, a smoothed transition from the linear to the nonlinear field is encountered in terms of load-displacement curve, since the nonlinear links progressively yield leading to an accurate simulation of the axial-flexural response. Precisely, by considering two generic panels $k$ and $l$ connected by an interface, with size in the direction orthogonal to the interface equal to $L_1$ and $L_2$, respectively, the calibration of a generic link with interspacing equal to $\lambda$ is obtained as the combination in series of two links, respectively associated to each of the two connected panels; in particular, by considering panel $k$, the stiffness of the link is equal to the axial stiffness of the relevant fiber with height equal to $L_1/2$; the tensile and compressive yielding strength $F_{ty1}$ and $F_{cy1}$ are calibrated considering the achievement of the tensile and compressive limit strengths in the fiber, and the corresponding ultimate displacements $u_{tu1}$ and $u_{cu1}$ are associated to the achievements of the ultimate tensile and compressive strains in the corresponding fiber.

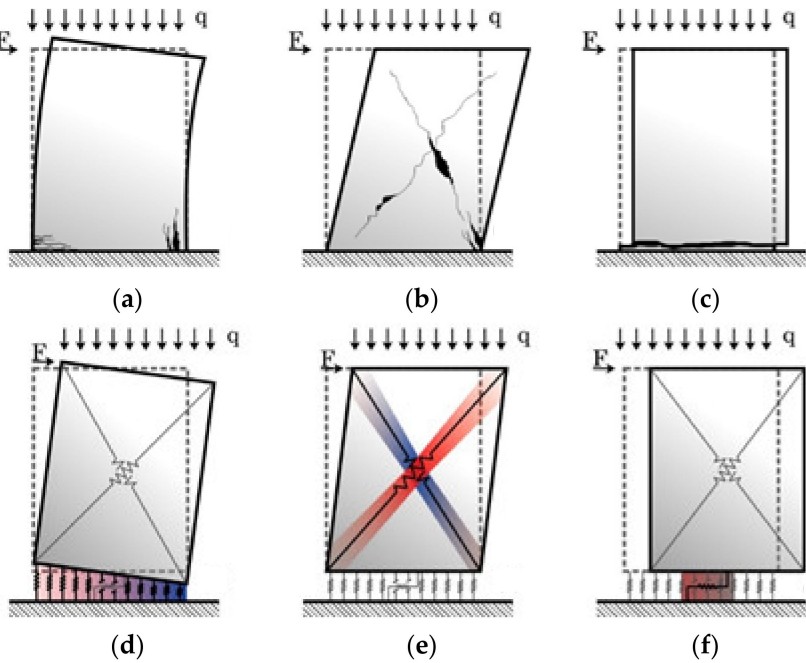

**Figure 1.** Main in-plane collapse mechanisms of masonry walls (**top**) and their simulation by means of the macro-element (**bottom**): (**a–d**) flexural failure; (**b–e**) shear-diagonal failure; and (**c–f**) shear sliding failure [15].

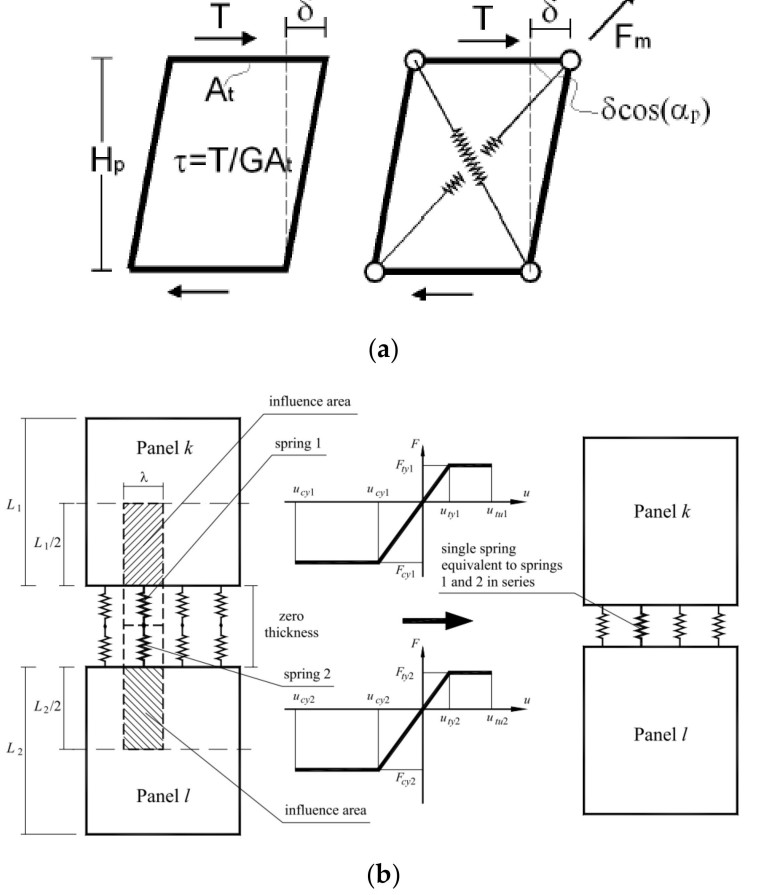

**Figure 2.** Calibration of the nonlinear links of the element: (**a**) diagonal and (**b**) flexural links [11].

The calibration of the diagonal links is based on an equivalence with the corresponding continuous media assumed to be shear deformable only. The initial linear elastic behaviour is therefore attributed to a diagonal link which is calibrated by assuming, in the hinged quadrilateral, the same drift as the continuous element possessing a shear stiffness $GA_t/H_p$, being $G$ the shear modulus, see Figure 2a. The inelastic response is ruled by a yielding domain generally associated with a Mohr-Coulomb or Turnsek and Cacovic constitutive law [17].

Although more complex constitutive laws can be combined with this approach (as proposed in [18,19]), here an elastic-perfectly plastic assumption is made for the axial-flexural behaviour, also in accordance with commonly adopted standards [20]. The DMEM here employed, and implemented in the software 3DMacro [21], can be adopted with a basic mesh (i.e., a single element is able to reproduce the nonlinear mechanical behaviour of an entire pier or spandrel) but also with a refined discretization when a more detailed response is desired. However, since each element is endowed with four degrees of freedom only, the needed computational effort is significantly reduced with respect to classical nonlinear FE approaches. On the other hand, with respect to other simplified approaches, based on frame elements, it presents meaningful advantages, since it avoids the introduction of rigid zones at the intersection between piers and spandrels, and also with reference to the interaction with steel or reinforced concrete frames. Precisely, the plane geometry of the element allows interacting along all the four edges either with other elements or with a contouring frame [12–14].

In Figure 3, a scheme highlighting the modelling of the interaction of a beam element with a masonry panel is depicted. The discrete interface effectively grasps the continuous interaction along the beam length with the masonry panel, and a single link governs the sliding interaction between the masonry panel and the frame. The beam element is endowed with external degrees of freedom at the two ends, and internal ones in correspondence of the internal node of each link; however, the internal degrees of freedom are condensed in the numerical procedure and do not contribute to increasing the size of the problem. The nonlinearities can independently occur in the masonry element, at the interfaces between masonry and the frame, and in the beam elements through the adopted lumped plasticity frame model. In particular, the inelastic behaviour of the plastic hinges can occur not only at the two beam ends but also along all its length, leading to a distributed plasticity beam. The yielding domain of the plastic hinge is evaluated according to the geometric and mechanical data of the cross sections; various models of plastic hinges can be adopted, namely neglecting the interaction between bending moment and axial force or, in the case of columns, evaluating a resistance domain able to account for the simultaneous presence of different forces. A further advantage of the geometrical consistency of the adopted model is the possibility of implementing a numerical model geometrically faithful to the actual structure, even in the presence of an irregular distribution of openings, thus avoiding any geometrical simplification.

Further upgrades of the model, indeed not exploited for the present study which is devoted to masonry buildings where the out-of-plane failure mechanisms are duly inhibited (i.e., the box behaviour of the structure is assumed), involved the extension to the three-dimensional behaviour, also in presence of curved geometry [22], which was achieved by introducing three additional out-of-plane degrees of freedom and conveniently extending the calibration procedures.

In this paper, the plane model is employed, i.e., no out-of-plane stiffness or resistance of the walls is accounted for, whilst the focus is kept on the interaction between the masonry panels and the steel frames in their own plane. In particular, even though the interaction of the three-dimensional element with beam elements was already considered in a previous work [23], the investigation here considered is limited to the beneficial effect of the frame to improve the in-plane behaviour of masonry walls by taking advantage of the presence of openings.

**Figure 3.** Qualitative representation of an interface between a beam column and the adjacent macro-elements and the corresponding degrees of freedom [12].

## 3. Validation of the Proposed Methodology

Aiming at validating the capability of DMEM for modelling steel frames around the opening of masonry walls, in this section an experimental test, already presented in the literature by Oña Vera et al. [2], is simulated. The experimental program was conceived considering the possibility of creating a new door opening in a solid brick masonry wall, and then introducing a steel frame contouring the new opening with the final goal of restoring the initial configuration in terms of structural performance. To this purpose, the actual construction process in case of new openings, from the cutting process to the retrofitting by means of steel frames, was simulated and the main steps have been numerically reproduced. Namely, the investigation presented in [2] included four walls corresponding to a blind wall, a wall with a door opening without any reinforcement, and two walls with an opening contoured by a steel frame, considering two different profiles, HEA140 and HEA 240. Such configurations were proposed for achieving a preliminary design of the frame for seismic actions by considering a detailed Finite Element model subjected to vertical loads and followed by in-plane cyclic lateral displacements histories. After the preliminary analyses the HEA140 section was adopted as design profile of the frame, subsequently tested in the experimental campaign, since it showed to be able to restore the resistance of the solid wall before the creation of the new opening. The layout of the tested specimen is depicted in Figure 4.

Although only the layout shown in Figure 4 was experimentally tested by Oña Vera et al., all the four configurations were numerically simulated [2], namely the solid wall (SW), the wall with opening (PW), the wall with opening and HEA140 steel frame (PW-HEA140), and the wall reinforced with HEA240 (PW-HEA240). Similarly, the numerical simulations of the four configurations have been performed in the present study by using the software 3DMacro [21], in which the DMEM is implemented. Since the numerical models performed by the authors in [2] provided results coherent with the experimental test, in this section the numerical simulations obtained by the DMEM are also compared with the numerical simulations reported in [2]. The numerical results are expressed in terms of capacity curves and failure modes.

Table 1 reports the set of mechanical parameters employed in the FEM simulations in [2], which have been also adopted for the simulations performed with the DMEM method discussed in the present research.

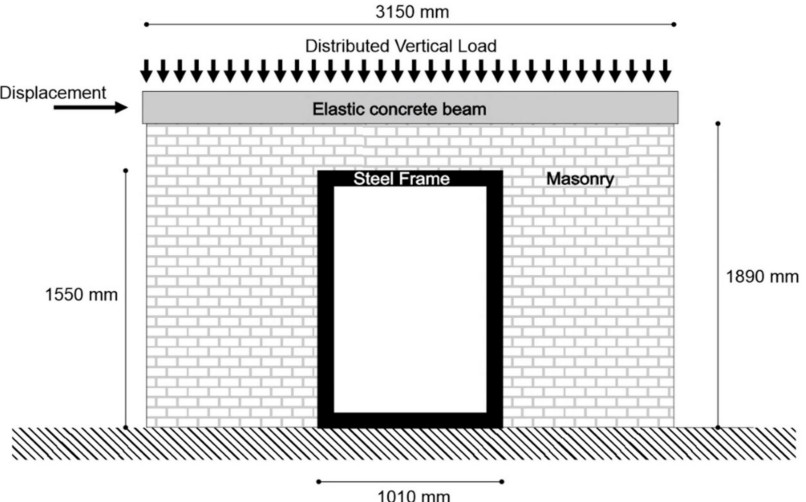

**Figure 4.** Geometric layout and load distribution of the tested prototype in [2].

**Table 1.** Masonry mechanical properties (adopted from [2]).

| Property | Symbol | Unit | Value |
|---|---|---|---|
| Elastic modulus | $E_M$ | MPa | 5344 |
| Poisson coefficient | $\nu$ | - | 0.2 |
| Mass density | $\gamma_m$ | kN/m$^3$ | 17 |
| Compressive strength | $f_m$ | MPa | 6.3 |
| Tensile strength | $f_t$ | MPa | 0.18 |
| Shear friction coefficient | $\mu$ | - | 0.78 |
| Cohesion | $c$ | MPa | 0.28 |
| Shear modulus | $G$ | MPa | 2227 |

In this research the implemented models were firstly subjected to the self-weight plus the additional distributed vertical load, and then to the lateral displacement controlled monotonic load history applied at the top of the structure.

A Mohr-Coulomb domain was assumed for the diagonal shear behaviour. Since in the experimental test no sliding was observed between frames and masonry [2], the sliding behaviour has been considered inhibited in the numerical simulations. Figure 5 reports the capacity curves and the relevant failure modes of the two unreinforced models. Figure 5a represents the failure mode of the solid wall (SW) specimen (modelled by means of 40 macro-elements), whereas Figure 5c refers to the wall with an opening (PW), which is characterized by 42 macro-elements. The corresponding capacity curves obtained by means of the DMEM approach are compared with those obtained through the FEM model [2] in Figure 5b,d, respectively. It is worth stressing that the FEM simulations are already reported in Oña Vera et al. [2] and are considered here as reference results.

In both the SW and PW configurations, the collapse mechanisms obtained through the DMEM are consistent with those reported in [2]. The capacity curves are in good agreement in terms of initial stiffness and peak load, whereas the post-peak behaviour shows some discrepancies. Precisely in the case of the SW, since the collapse mechanism is governed by the flexural behaviour, the simple constitutive law here adopted, does not lead to the smoothed softening observed in the FEM simulations. In the PW configurations, since the diagonal shear behaviour is also involved, some drops in the global resistance are observed as the ultimate drift is achieved in some panels, whereas the smoothed post peak softening behaviour is observed in the case of the smeared crack FEM numerical simulations.

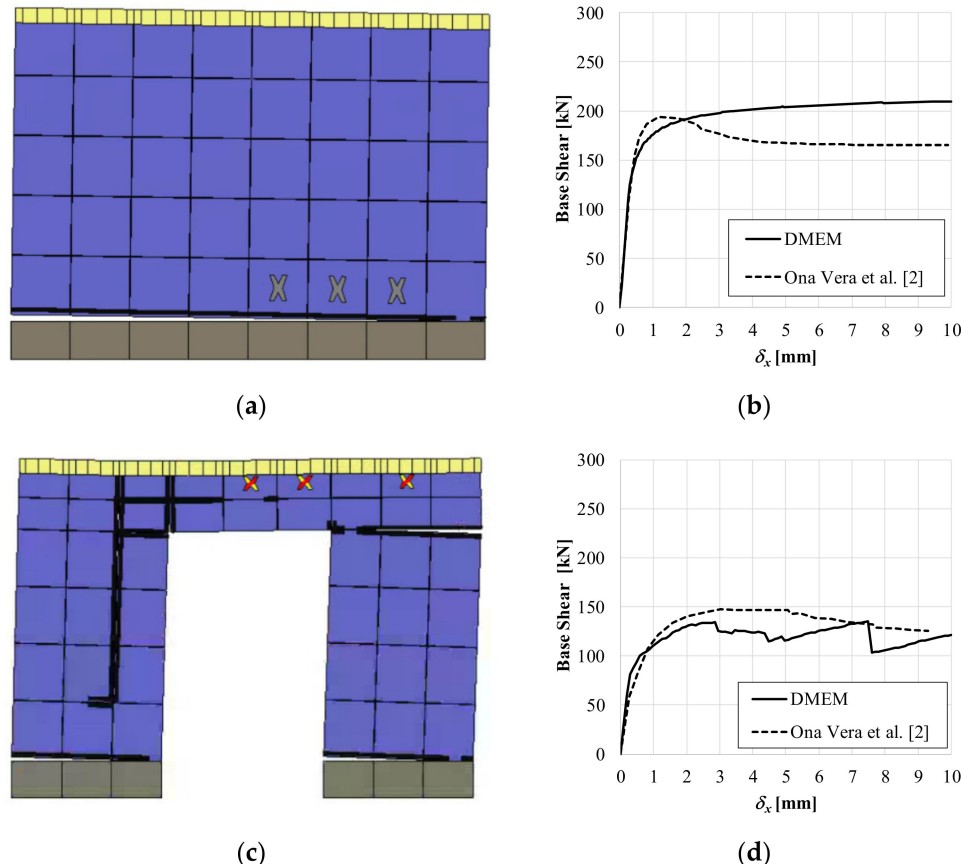

**Figure 5.** Comparison in case of solid wall (SW) and wall with openings (PW) in terms of (**a**,**c**) collapse mechanism and (**b**,**d**) capacity curves, respectively.

Figure 6 reports the comparisons of the wall with the opening surrounded by the steel frame. The numerical models are characterized by 61 macro-elements and 11 beam elements. As the figure shows, although the DMEM is computationally less demanding, it is able to predict the global behaviour in terms of initial stiffness, peak load, and softening branch. Consistently with the numerical and experimental results presented in [2], the presence of the steel frame is able to guide the collapse mechanism of masonry piers towards the more dissipative failure mode of the diagonal shear cracking, which leads to a significant increase of the global peak load (about 150 kN for the PW configuration, about 200 kN for the PW-HEA140 configuration and about 250 kN for the PW-HEA240 configuration). Minor differences between DMEM and FEM approaches can be observed in terms of global ductility in the case of the PW-HEA240 configurations, since a more pronounced softening is encountered in the case of the Finite Element model. It is worth mentioning that, although the adoption of a more resistant cross section (PW-HEA240) leads to an increase of the global resistance, it turns out that the global ductility is reduced with respect to the configuration with a weaker steel profile (PW-HEA140); this implies that strongest steel profiles do not necessary lead to a better global behaviour, and the retrofitting design must be well balanced in terms of strength and ductility.

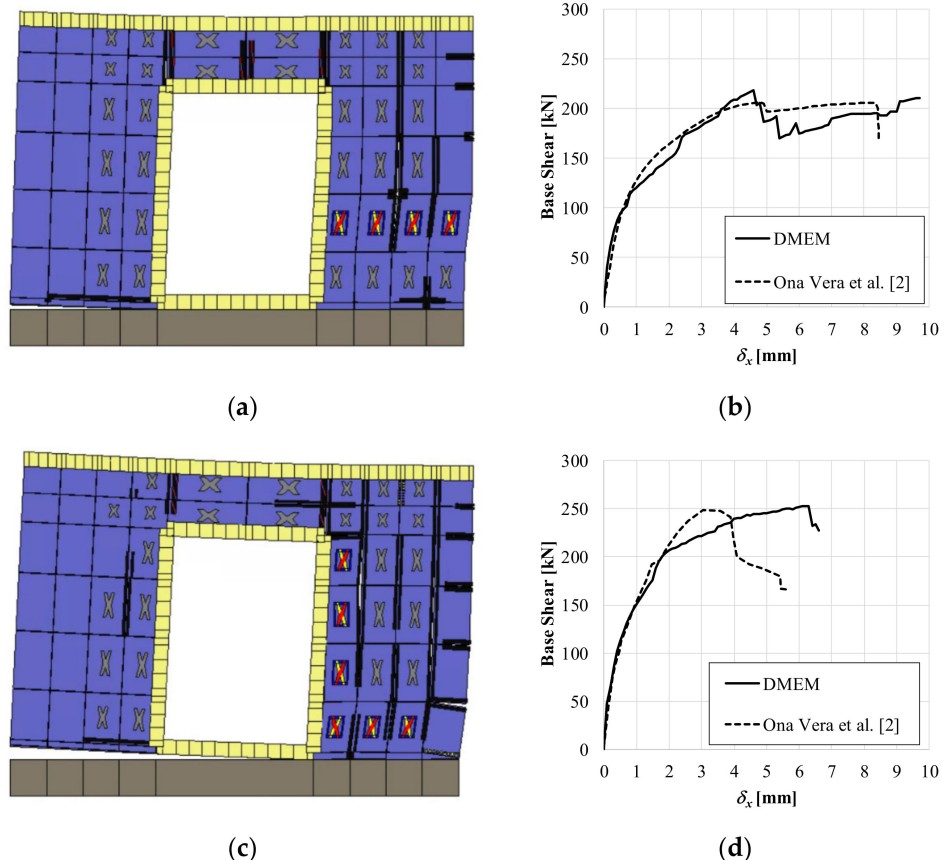

**Figure 6.** Comparison in case of wall with an opening ringed with a HEA140 steel frame PW-HEA140 and with a HEA240 steel frame PW-HEA240 in terms of (**a**,**c**) collapse mechanism and (**b**,**d**) capacity curves, respectively.

## 4. Application to a Full-Scale Benchmark Structure and Discussion

Aiming at applying the proposed low-impact retrofitted strategy to a case study that has been already investigated in the literature without considering any retrofitting strategies, the *via Martoglio* masonry wall benchmark representative of an URM multi-storey building, placed in the city of Catania, Italy [15], has been considered. The wall, made by regular unit masonry bricks, is characterised by a 300 mm thickness, except at the last level where the thickness is 160 mm. The 300 mm thick dimension is related to two wythes of interlocked brick layers covered by the external plaster layers while the 160 mm dimension identifies the total thickness of a single-wythe plastered brick wall. Nonlinear concrete edge beams are placed at each floor except the roof. These in-depth concrete edge beams are 24 cm in height reinforced by 4Ø12 longitudinal bars and Ø6 at 25 cm stirrups, uniformly distributed. The edge beams are built with concrete with Young's modulus $E$ = 28,821 MPa, compressive strength $f_c$ = 24.6 MPa, and tensile strength $f_t$ = 2.17 MPa. The reinforced bars have a yield limit strength $f_y$ = 335 MPa.

A symmetric arrangement of openings (Figure 7) defines the structural layout. As the figure shows, all the openings, except the large central door placed at the ground level, have been encircled with steel frames. The use of two different steel cross sections has been considered. Specifically, the EU sections IPE 100 and HE100A have been adopted. The steel material that has been considered for the frame is S235 type, largely adopted for this type of steel elements.

Three configurations have been analysed and compared in terms of capacity curves and collapse mechanisms:

- Configuration 1—URM wall, characterised by uniform mechanical properties reported in Table 2 with nonlinear floor edge beams at each level except the last one.
- Configuration 2—Configuration 1 where all the openings are encircled with IPE100 steel frames.
- Configuration 3—Configuration 1 where all the openings are encircled with HE100A steel frames

The two selected cross sections, IPE100 and HE100A, have weight equal to 8.1 kg/m and 16.7 kg/m, respectively and both can be considered as a low-cost retrofitting strategy.

As deeply investigated in [15] the multilevel wall peak base shear forces are influenced by the presence of concrete edge beams. In [15] three different configurations are compared: (i) the URM wall, (ii) the wall with linear concrete edge beams and, lastly, (iii) with non-linear concrete edge beams (Configuration 1 in this paper), by considering several computational approaches (nonlinear limit analysis, DMEM, continuum nonlinear FEM and high fidelity nonlinear FEM micro-modelling). The URM configuration, previously investigated by other authors in [24–27], is additionally reported in Figure 8 in terms of capacity curve for sake of completeness. As discussed in [15] the selected numerical approaches lead to a satisfactory agreement between the results provided by different numerical strategies and all the approaches confirm that the assumption of nonlinear concrete edge beams lead to realistic increment of the base shear peak force.

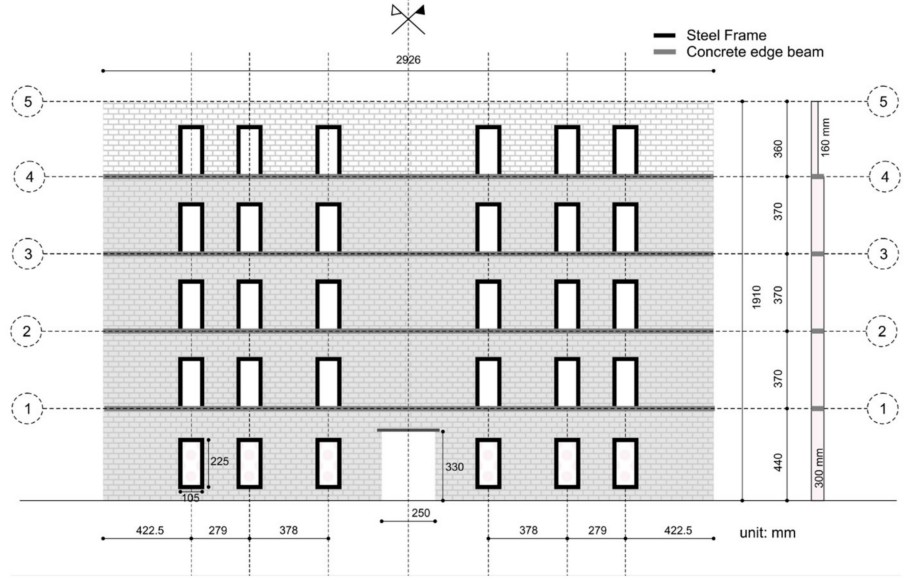

**Figure 7.** Geometrical layout of the multi-storey masonry wall.

**Table 2.** Resistance parameters adopted in the model—masonry (from [15]).

| Property | Symbol | Unit | Value |
|---|---|---|---|
| Elastic modulus | $E_m$ | MPa | 1600 |
| Compressive strength | $f_m$ | MPa | 6 |
| Tensile strength | $f_t$ | MPa | 0.24 |
| Tensile ductility | | - | 1.05 |
| Compressive ductility | | - | $\infty$ |
| Shear modulus | $G$ | MPa | 540 |
| Shear friction coefficient | $\mu$ | - | 0.5 |
| Shear strength | $\tau_0$ | MPa | 0.16 |
| Mass density | $\gamma_m$ | kN/m$^3$ | 17 |

Figure 8 reports the capacity curves of the three configurations. The lower dotted line represents the behaviour of the URM wall. As the figure shows, when the frames are applied the walls exhibit conspicuous strength increments. In this benchmark, the

peak force value increases from 1780 kN (Configuration 1) up to 2980 kN and 2860 kN for Configuration 2 and 3, respectively. As already underlined in Section 3, stiffer cross sections may lead to lower global ductility. In this case Configuration 2 is denoted by a slightly lower peak force in comparison to Configuration 3, but an almost similar post peak branch.

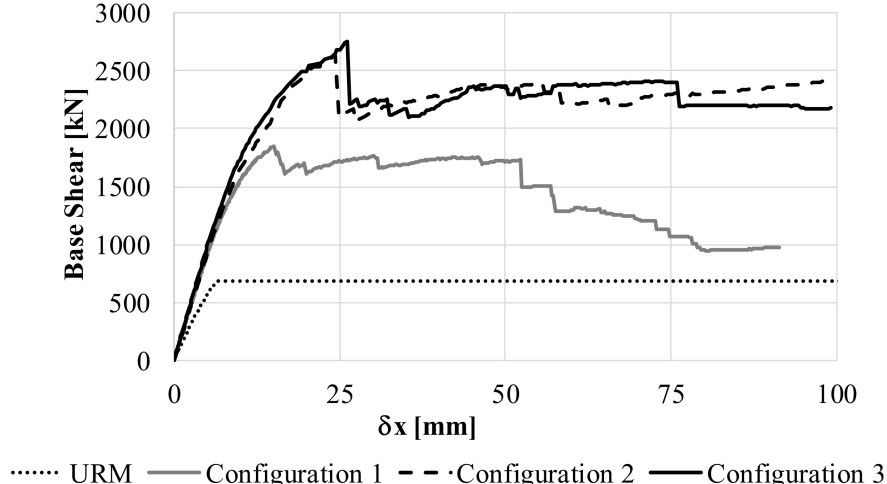

**Figure 8.** Capacity curves.

Figure 9 reports the collapse mechanisms of the three configurations. Both the retrofitted models denote spread failures in the piers and spandrels at different levels, contrarily to Configuration 1 in which the failures are mainly localized in the spandrels. Due to the presence of the ductile steel frames, the failure mechanisms involve piers and spandrels panels. Some panels that exhibited flexural failure mechanisms in Configuration 1 show a diagonal shear collapse mode in the retrofitted layouts. Although the shear failures are generally defined by lower global ductile behaviour, the confinement effects due to the steel frames and their inelastic response allow the obtaining of a more ductile global mechanism (see Configuration 2 in Figure 8) characterized by different dissipation sources that will guarantee a better behavior particularly under dynamic loadings.

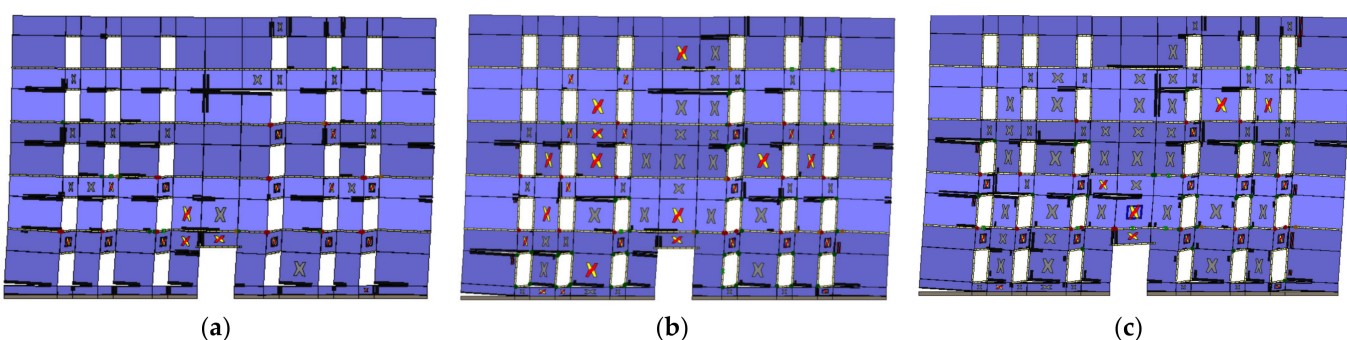

**Figure 9.** Collapse mechanisms of Configuration (**a**) 1, (**b**) 2 and (**c**) 3.

As already emphasized, this strategy allows for plastic hinges in the steel frames which increase the hysteretic dissipated energy. Figure 10 reports the detail of the ground level of Configuration 2 in which the green points indicate the plastic hinges position.

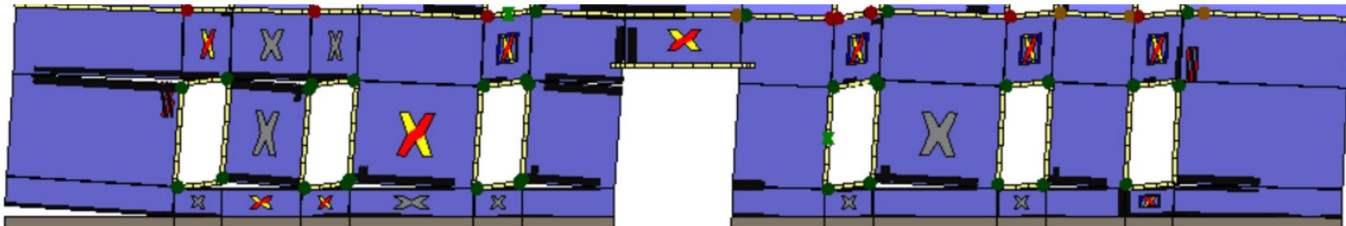

**Figure 10.** Detail of collapse mechanism in Configuration 2.

The numerical simulations here reported suggest that the DMEM can be successfully applied for modelling low-impact retrofitting strategies based on the introduction of dissipative steel frames in the openings of URM masonry walls.

## 5. Conclusions

In this paper the adoption of steel frames along the perimeter of existing openings in URM buildings is adopted and numerically simulated as a low-impact retrofitting measure. The adopted strategy, already proposed by Oña Vera et al. in [2], allows a fast installation and can be combined with other energy efficient design solutions. The efficacy of the seismic retrofitting strategy has been evaluated by means of an efficient Discrete Macro-Element Method (DMEM) able to account for the continuous interaction between the steel frames and the surrounding masonry panels. The adopted DMEM, being characterized by a low computational burden, can be applied for the numerical simulations of entire buildings with computational effort compatible with practical engineering applications. In the paper, the adopted numerical approach has been firstly validated with an experimental test conducted on a simple benchmark prototype by Oña Vera et al. [2], and then applied to a benchmark representative of a typical URM masonry multi-storey building to investigate the performance of the proposed low-impact seismic retrofitting solution at the building scale. The adoption of steel frames in existing openings to retrofit existing masonry structures implies a complex beneficial interaction with masonry piers and spandrels. The frame around the openings, besides increasing the strength and ductility of the combined system, introduces a confining effect in the masonry piers and spandrels that leads to a more dissipative collapse mechanism. The obtained results have shown that the proposed retrofitting strategy can successfully be applied as a low-cost, low-impact retrofitting strategy providing an increment of the global strength as well as of the global ductility. The results have shown that the introduction of dissipative steel frames around the existing openings can be a very useful and powerful low-impact retrofitting strategy that can be also adopted in combination with other retrofitting measures. Since very little research has been conducted so far for evaluating this cost-effective retrofitting solution, many aspects can be further investigated, such as the influence of the steel frame in the out of plane behavior of the masonry wall, the identification of an optimal shape of the frame for increasing the energy dissipations etc. These aspects could be the subject of future research.

**Author Contributions:** Conceptualization, I.C. and G.O.; writing—original draft preparation, F.C.; writing—review and editing, I.C. and S.C. All authors have read and agreed to the published version of the manuscript.

**Funding:** This research was funded by *Progetto DPC-ReLUIS 2019-2021 WP5: Interventi di rapida esecuzione a basso impatto ed integrati*.

**Conflicts of Interest:** The authors declare no conflict of interest.

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
