# Peer review of "A Discrete Macro Element Method for Modelling Ductile Steel Frames around the Openings of URM Buildings as Low Impact Retrofitting Strategy"

_sustainability, doi:10.3390/su13179787_

Round 1
Reviewer 1 Report
- The abstract was written too long. The research objective, scope, method, and results should be briefly summarized.
- The analytical model (methodology) is not clear and should be described step by step. Also, the limitation of model should be discussed clearly and that how the results might be used in practice.
- More information on the analytical model for validation, including the composition of the model and material, the method of applying load or displacement.
- A detailed explanation of Figure 5 (b) is required.
- Opening is confirmed in the Fig. 5, how was the related factor handled in the analysis?
- In chapter 4, a figure that can confirm the configuration of the model is required.
Author Response
The authors wish to thank the reviewer for his/her comments. In the following a detailed point-by-point response is reported.

Reviewer 2 Report
Please, see the attached file.

Author Response
The authors wish to thank the reviewer for having appreciated the paper and for all the comments and suggestions aimed at improving the final version of the manuscript. In the following a detailed response to each suggested point is reported.

Reviewer 3 Report
This manuscript aims to present a numerical investigation on the use of low seismic retrofitting using steel elements to contour the opening of URM building structures. This study can be promising for the construction industry. The authors are advised to recheck the manuscript carefully to avoid English grammar errors.
I believe that that this research work is interesting. However, some revisions and minor clarifications are required before the manuscript is suitable for publication, as outlined in the following points.
Abstract:\
- Lines 9-13: Please revise the sentence.
- “within the ReLUIS 2019-2021 project” or “within a ReLUIS 2019-2021 project”?
- The authors do no provide solid and summary conclusions obtained in this investigation. Please revise it.
Section 1
- The state-of-art review is not sufficient. It is not clear the advantages and disadvantages of each retrofitting technique. Some important research works are missing (i.e. DOI: 10.1016/j.conbuildmat.2020.120520). It would be interesting to provide some technical findings observed in experimental tests.
- The novelty of this study needs to be improved. It is not clear for the reader the methodology adopted.
- No discussion about costs is provided. This aspect would be interesting also to increment the value of this literature review
Section 2
- Figure 2: There are several parameters in this Figure not introduced in the text. Please revise it.
- Besides the OOP behaviour is not considered in the numerical model, it would be interesting for the reader to discuss the possible implications of this simplification.
Section 3
- The authors need to provide more details concerning the test results (e.g. damages observed and force-displacement results)
- Figure 4: units are missing.
- Table 1: These values were defined based on the results of material testing or were assumed? Please clarify it.
- Figure 5: Please use the same vertical units. Some small decays can be observed in the top of the numerical response (bottom plot). Why did this occur? It seems that the numerical model overestimates the initial stiffness and maximum strenght.
- Is this model able to simulate cyclic loading?
Section 4
- Please justify the values adopted in Table 2.
- The results discussion is poor. Please improve it.
Section 5
- The authors should suggest future works
Author Response
The authors wish to thank the reviewer for appreciating the paper and for the suggested revisions. In the following a detailed response to each suggested point is reported. Furthermore, the text has been fully reviewed aiming to emphasise the aspects highlighted by the reviewer. The text has been carefully checked and the English grammar errors have been amended.

Round 2
Reviewer 1 Report
The manuscript has been well modified in consideration of the comments.
Author Response
Authors’ Response to Reviewers of:
A Discrete Macro Element Method for modelling ductile steel frames around the openings of URM buildings as low impact retrofitting strategy
Giuseppe Occhipinti*, Francesco Cannizzaro, Salvatore Caddemi and Ivo Caliò
The Authors would like to thank the Reviewers for all their comments that helped the Authors to increase the quality of the manuscript. All the relevant changes of the second revision have been green highlighted in the paper and described in this document. Detailed answers to each point are provided below with attention to the critical issues addressed by the reviewers.
Reviewer #1: Review of the paper:
The manuscript has been well modified in consideration of the comments.
The authors wish to thank the reviewer for his/her final comments.

Reviewer 2 Report
The reviewer appreciates the effort of the authors to address the reviewer’s comments and acknowledges that most changes in the revised version of the manuscript added quality and depth to the paper. The revised manuscript presents a numerical scheme using Discrete Macro-Element Modelling (DMEM) and investigates the efficiency of the adopted low-impact strengthening solution.
After carefully reading the revised manuscript, the reviewer considers that it still presents some limitations that need to be addressed before the manuscript is considered for publication. Specifically:
Comments
- Abstract – Conclusions: The revised manuscript still mentions that it “proposes” the use of steel frames around openings in URM buildings. However, the manuscript “adopts” the already existed retrofit methodology [2]. Therefore, the authors are kindly requested to replace “proposes” (or “proposed”) in the abstract and the conclusions of the revised manuscript with “adopts” (or “adopted”, respectively).
- Figures 5, 6: Please explain the higher elastic stiffness that characterizes the proposed DMEM model when compared to the numerical results of [2], which already show a higher stiffness compared to the experimental results (as shown in Figure 13 in [2]). In other words, the results obtained from the proposed DMEM model instead of converging to the experimental results of [2], they diverge from them. Please, elaborate on why this happens in the revised manuscript.
- Figure 5: In accordance with my previous comment #2, no discussion is presented on the results of Figure 5. A mere presentation of the two curves is not sufficient. A more detailed justification of the similarities and/or discrepancies between the two curves, especially when compared with experimentally derived curves, is necessary.
- Section 4 (Figure 8): The most important comment of my initial review has not been addressed. Section 4 in the revised manuscript remained unchanged (compared to the corresponding Section 4 of the original manuscript). Specifically, Figure 8 considerably differs from the corresponding capacity curves of the same case study published in the literature. Unfortunately, no effort has been made to address these discrepancies in the revised manuscript. On the contrary, the whole discussion (at the authors’ reply document) led to their under review study [15] by promoting the analyses conducted in Ref. [15] rather than in the present manuscript. The justifications behind the discrepancies of Figure 8 need to be addressed in the current revised manuscript and not in Ref. [15]. Please, revise the present manuscript accordingly.
Author Response
Authors’ Response to Reviewers of:
A Discrete Macro Element Method for modelling ductile steel frames around the openings of URM buildings as low impact retrofitting strategy
Giuseppe Occhipinti*, Francesco Cannizzaro, Salvatore Caddemi and Ivo Caliò
The Authors would like to thank the Reviewers for all their comments that helped the Authors to increase the quality of the manuscript. All the relevant changes of the second revision have been green highlighted in the paper and described in this document. Detailed answers to each point are provided below with attention to the critical issues addressed by the reviewers.
Reviewer #2:
The reviewer appreciates the effort of the authors to address the reviewer’s comments and acknowledges that most changes in the revised version of the manuscript added quality and depth to the paper. The revised manuscript presents a numerical scheme using Discrete Macro-Element Modelling (DMEM) and investigates the efficiency of the adopted low-impact strengthening solution.
The authors wish to thank the reviewer for appreciating the improvement to the manuscript quality. In the following the reviewer’s suggestions are addressed aiming at fully covering all possible limitations.
After carefully reading the revised manuscript, the reviewer considers that it still presents some limitations that need to be addressed before the manuscript is considered for publication. Specifically:
Comment #1. Abstract – Conclusions: The revised manuscript still mentions that it “proposes” the use of steel frames around openings in URM buildings. However, the manuscript “adopts” the already existed retrofit methodology [2]. Therefore, the authors are kindly requested to replace “proposes” (or “proposed”) in the abstract and the conclusions of the revised manuscript with “adopts” (or “adopted”, respectively).
RESPONSE. As requested, the verb ‘to propose’ has been replaced with the verb ‘to adopt’ in order to emphasize that the retrofit methodology has already been proposed.
Comment #2 Figures 5, 6: Please explain the higher elastic stiffness that characterizes the proposed DMEM model when compared to the numerical results of [2], which already show a higher stiffness compared to the experimental results (as shown in Figure 13 in [2]). In other words, the results obtained from the proposed DMEM model instead of converging to the experimental results of [2], they diverge from them. Please, elaborate on why this happens in the revised manuscript.
RESPONSE. The authors thank the reviewer for giving the opportunity of further checking the differences between the results. In the previous version the EMy [2] was considered. In the new version the value has been updated to EMx [2] consistently to the elastic modulus adopted in the numerical simulations performed in the same study [2]. Consequently, table 1 has been amended and all the figures have been updated. As the new figures show, the initial tangent stiffness agrees with the reference curves. The authors are grateful to the reviewer for all the constructive comments which allowed improving the match with the results already available in the literature.
Comment #3 Figure 5: In accordance with my previous comment #2, no discussion is presented on the results of Figure 5. A mere presentation of the two curves is not sufficient. A more detailed justification of the similarities and/or discrepancies between the two curves, especially when compared with experimentally derived curves, is necessary.
RESPONSE. The manuscript has been improved with additional comments considering the Comments 2 and 3, better highlighting analogies and discrepancies with the results available in the literature.
Comment #4 Section 4 (Figure 8): The most important comment of my initial review has not been addressed. Section 4 in the revised manuscript remained unchanged (compared to the corresponding Section 4 of the original manuscript). Specifically, Figure 8 considerably differs from the corresponding capacity curves of the same case study published in the literature. Unfortunately, no effort has been made to address these discrepancies in the revised manuscript. On the contrary, the whole discussion (at the authors’ reply document) led to their under review study [15] by promoting the analyses conducted in Ref. [15] rather than in the present manuscript. The justifications behind the discrepancies of Figure 8 need to be addressed in the current revised manuscript and not in Ref. [15]. Please, revise the present manuscript accordingly.
RESPONSE
Aiming at better fitting the reviewer’s suggestions in the revised version of the manuscript we introduced further changes. Namely, Figure 8 reports a further push-over curve reporting the results obtained with the DMEM approach considering the URM configuration (i.e., without the presence of floor beams), which was previously investigated by other authors (see references [24-27] of the revised version of the manuscript). The authors are confident that the additional curve may represent a starting point to show the reliability of the model to analyse both the reference configuration (in the presence of nonlinear floor beams) and the retrofitted one. Precisely, it can be observed how the URM configuration exhibits a global resistance internal to the range suggested by the reviewer in the previous round of comments (600-1200 kN). The authors are also sure that the reviewer will understand that further details on the URM configuration might fall outside the scope of the present study which is focused on the benefits provided by the adopted retrofitting strategy already suggested also by other authors, as in reference [2].

Reviewer 3 Report
The authors carried out all the revisions and recommendations made by the reviewers. The quality of the manuscript was improved and it is now suitable for publication.
Author Response
Authors’ Response to Reviewers of:
A Discrete Macro Element Method for modelling ductile steel frames around the openings of URM buildings as low impact retrofitting strategy
Giuseppe Occhipinti*, Francesco Cannizzaro, Salvatore Caddemi and Ivo Caliò
The Authors would like to thank the Reviewers for all their comments that helped the Authors to increase the quality of the manuscript. All the relevant changes of the second revision have been green highlighted in the paper and described in this document. Detailed answers to each point are provided below with attention to the critical issues addressed by the reviewers.
Reviewer #3:
The authors carried out all the revisions and recommendations made by the reviewers. The quality of the manuscript was improved and it is now suitable for publication.
The authors wish to thank the reviewer for all his/her the revisions and for appreciating the quality of the manuscript.
